# Analytic Model of Maximal Experimental Value of Stress Intensity Factor K_Q_ for AA2519–AA1050–Ti6Al4V Layered Material

**DOI:** 10.3390/ma13194439

**Published:** 2020-10-06

**Authors:** Maciej Kotyk

**Affiliations:** Faculty of Mechanical Engineering, University of Science and Technology in Bydgoszcz, 85-796 Bydgoszcz, Poland; mackot001@utp.edu.pl; Tel.: +48-523-408-212

**Keywords:** fracture mechanics, cryogenic conditions, lightweight materials, aluminum alloy, titanium alloy, explosive welding, stress intensity factor modeling

## Abstract

The article presents the results of the author’s tests involving the determination of the maximal experimental value of the stress intensity factor K_Q_. This value was determined for a layered material obtained as the result of explosive welding of three alloys: AA2519, Ti6Al4V and AA1050, and separately for each material. In both cases tests were conducted for two temperatures—the ambient temperature (293 K), and cryogenic temperature (77 K). A model for initial assessment of the K_Q_ value of AA2519–AA1050–Ti6Al4V (Al–Ti) layered material has also been presented. The proposed model has been developed so as to enable the determination of the curve course of load–COD for Al–Ti layered material using nominal stresses defined on the basis of a real load–COD course, obtained for the base materials, for both temperature conditions.

## 1. Introduction

Layered materials are being more and more frequently used in the construction of engineering structures. This is due to at least two factors. The first is associated with a decrease in the object mass without losing the mechanical properties of the structure’s top layer. Naturally, the mechanical properties of the internal layer material differ from the external ones though they are characterized by lower density. The proposed solution can be applied in objects that are exposed to an aggressive environment [1,2,3,4,5,6,7]. The second advantage is the construction cost reduction, which is significant from the point of view of investors. In this type of structure, a material whose purchase costs are lower plays the role of a filler, medium or stiffening. This is the case, for example, in installations where a liquid, which is aggressive to carbon steels, is transported. In this case, a material that is resistant to the harmful effects of the liquid is placed on the aggressive side and carbon steel on the outside. As described by Blazynski in his book [1], explosive welding is also used to join materials intended for the structures described. Although explosive welding is expensive, it is ultimately cheaper to make an installation from a sandwich material than to make it entirely from a material that is resistant to the aggressive influence of the transported fluid.

One of the methods for joining materials with significantly different mechanical properties is explosive welding. Unlike low energy methods, explosive welding provides the possibility of bonding materials which cannot be connected by standard methods. These composites include, e.g., stainless steel and copper [8], stainless steel and austenite steel [9], or layered materials built from alloys of aluminum and titanium [10,11,12,13,14,15].

An explosive welded layered material made from three alloys with significantly different mechanical properties is an example of such a material. These materials are AA2519, AA1050 and Ti6Al4V. The mechanical characteristics of these alloys used as base materials are known and widely described in the literature [11,14,15,16,17,18]. However, there are not many scientific reports regarding a composite explosively welded from these materials. This is because the described bimetal is not commonly used and its mechanical properties still need to be determined experimentally. Only a few works can be found in the publicly available database of magazines, provides information about selected mechanical properties of explosively welded laminated material AA2519–AA1050–Ti6Al4V. The studies that are available include, among others, works [16,17], which provide results of the determination of the tensile properties for the discussed material, and [18,19,20], which provide information on the micro toughness, microstructure and micro phenomena that occur in the material under the influence of mechanical factors. During the analysis of literature on the subject, articles were also encountered in which an attempt was made to determine the fatigue life of the described layered material. These are papers [20,21,22,23], which address the fatigue life of Al–Ti material. Additionally, the article [22] presents the analysis of microstructure of fracture samples made of Al–Ti material.

This study covers two aspects. The first includes selected characteristics of the layered material and base materials, the research equipment and selected mechanical properties of the tested materials. A special focus is put on quantitatively characterizing the fracture toughness of the tested materials.

Each of the base materials used to produce Al–Ti layered material is characterized by individual features describing its fracture toughness. After explosive welding a completely new material with different mechanical characteristics is created. The question has arisen whether it is possible to estimate the value of this characteristic for the layer material, resulting from explosive welding of these materials, on the basis of known characteristics determined for base materials. The second aspect addressed in this paper is an attempt to answer this question. This part of the article is a presentation and verification of an analytic model which was assumed to be a tool allowing an initial assessment of the K_Q_ value for a layered material made of base materials whose K_Q_ value is known.

The main goal of this study is to compare an experimentally obtained crack opening diagram, and thus the maximal experimental K_Q_ value for the Al–Ti layered material, with the one obtained by means of an analytic model.

The value of the maximal experimental stress intensity factor has already been determined in tests for both ambient and cryogenic conditions. The described characteristics were determined according to the equation:(1)$$K_{Q}=\left(\frac{P_{Q}}{BW^{\frac{1}{2}}}\right)\cdot f\left(\frac{a}{W}\right)$$
where:(2)$$f\left(\frac{a}{W}\right)=\frac{\left(2+\frac{a}{W}\right)\left(0.886+\frac{4.64a}{W}-\frac{13.32a^{2}}{W^{2}}+\frac{14.72a^{3}}{W^{3}}-5.6a^{4}\right)}{{\left(1-\frac{a}{W}\right)}^{\frac{3}{2}}}$$
where *P_Q_*—load determined on the basis of the experimental curve, *B*—sample thickness, *W*—nominal sample width and *a*—crack length.

Details of those tests can be found in Boroński et al. [24]. However, it needs to be stressed that the tests described were focused on the influence of delamination between particular layers of the Al–Ti material on the mechanical quantities characterizing the material fracture toughness.

The article shown contains an attempt to describe the strength of the layer connecting the base materials, already after the explosive welding. The main problem encountered during this attempt to describe the strength of this layer is the delamination occurring between the materials.

However, this article uses previous work to propose an analytical model, based on the information contained therein, which would allow to determine the stress intensity factor for Al–Ti material on the basis of the same amount but determined for base materials. It should be noted that the presented model strongly depends on the geometry of the sample.

## 2. Experimental

### 2.1. Base Materials 

The base materials used for manufacture of the Al–Ti composite include two aluminum alloys. One of them is AA2519 alloy. This is a new material which has not been explicitly described in the literature. One of its main advantages is its outstanding resistance to adverse weather and environmental conditions such as saline sea water. Furthermore, the alloy is also resistant to stress-related corrosion. It is referred to as hard to weld, which complicates some technological production and montage processes.

The characteristics listed above make the material applicable in military vehicles. It is used for covering external layers of military amphibious vehicles, reducing their mass and providing them with corrosion and ballistic resistance. The material gained popularity as soon as its plastic treatment technology had been developed, and then it stopped being used only in the form of sheet metals and plates [25,26,27]. See Table 1 for information about the material’s chemical composition.

Ti6Al4V titanium alloy (Table 1), is a construction material that is popular due to its low density and high strength as compared to steel materials, and it is widely described in the literature in terms of both its mechanical characteristics [14,28,29] and subtractive technologies [30]. The mechanical properties of the above-mentioned alloy are attributed to its crystalline structure composed of α + β phases, where the coarse grain phase α occurs along with β phases saturated with vanadium and aluminum with locations on the grain boundaries. AA1050 alloy is used in the electric and electronic industries due to its electrical conductivity. It can also be found in the construction industry thanks to its thermal conductivity. This advantage, in combination with very good plasticity and susceptibility to plastic treatment, made AA1050 a component material of a layered material consisting of AA2519–AA1050–Ti6Al4V. Its small thickness and above-mentioned plasticity are thought to play the role of a vibration silencer to reduce the brittleness of the transition zone between the layers of titanium and aluminum. Additionally, the chemical composition of the base materials that make up the analyzed composite is presented in Table 1.

A series of experiments involving the determination of the static properties of the main base materials, that is, the AA2519 and Ti6Al4V alloys, were conducted prior to fracture mechanics tests. The results and a detailed description of the tests including cryogenic conditions can be found in Boroński et al. [16]

### 2.2. Layered Material

The main research object is an explosive welded material which consists of the three alloys: AA2519–AA1050–Ti6Al4V. According to the developed technology, the welding of these materials involved parallel plating of AA2519 aluminum and Ti6Al4V titanium layers, whereas the lightweight alloy was a layer to be applied. As a result of this, the titanium had no direct contact with the explosion. Prior to the welding process, the aluminum layer was rolled onto a 0.2 mm thick layer of AA1050 alloy. This procedure was in order to create a buffer layer between the materials. The process of welding was carried out in such a way that the speed of the flame front during detonation (V_k_) ranged from 1850 m·s^−1^ to 2000 m·s^−1^, with the approach angle equal to 15°, whereas the impact zone shift speed (V_c_) for the analyzed material was about 500 m·s^−1^. A schematic presentation of this process is given in Figure 1.

The effect of the above-described procedures is the achievement of a speed at the level of 420–620 m·s^−1^, and the occurrence of a geometrically complicated structure in the bonding layer, which is discussed in detail in Bazarnik et al. [17]. The described material right after welding is illustrated in Figure 2.

### 2.3. Test Stand

The major element of the stand used for tests of both the static properties and fracture toughness was a hydraulic strength testing machine Instron 8501 with a system for the recording of three measurement channels, that is, the specimen loading force, the machine piston displacement, and the crack opening measured by an extensometer. The strength testing machine is presented in Figure 3. 

Experimental tests were conducted for both ambient and cryogenic conditions, which should be understood as the immersion of a specimen in a liquid nitrogen bath. To provide an identical temperature throughout the experiment, the tested specimen was placed in an environmental chamber whose lower pin was fixed directly on the strength testing machine. The chamber was made from two layers of Inconel with elements of thermal insulation placed between them, whereas the cover was made of Teflon to avoid freezing. The described element of the test stand is shown in Figure 3b,c.

As shown in Figure 3, acA4024-8gm GigE digital cameras (Basler, Ahrensburg, Germany) were placed on both sides of the test specimen. These devices allow you to take pictures delivered at eight frames per second at 12.2 MP resolution. It should be noted that a custom VS-TCT1-65/S PDE1telemetric lens (VS Technology Corporation, Tokyo, Japan) is mounted on the camera. Optical equipment was used for observation of the crack growth. The software recognized the crack growth by means of analysis of the displacement fields around the crack tip, using the image digital correlation method. This involves an analysis of two images of the object recorded before and after its loading. This is possible through determination of the differences in the form of the displacement of characteristic points located in the recorded reference image. Determination of the described displacement distribution is carried out by means of an analysis of the sum of the square differences in intensity of the images recorded for the two different stages of loading.

From a fragment of the analyzed image, being the environment of a selected point of the P(*x, y*) surface, whose size is given in pixels with dimensions *x* and *y*, sum *S* is calculated, which can be described as:(3)$$S=\;\overset{\beta }{\underset{i=\alpha }{\sum }}\overset{\zeta }{\underset{j=\gamma }{\sum }}{\left[I_{n}\left(x_{i},y_{j}\right)-I_{m}\left(x_{i},y_{j}\right)\right]}^{2}$$
where *α*, *β*—initial and final value of *x* coordinate index of the correlated images (*β* − *α* = *w*), *γ*, *ζ*—initial and final value of *y* coordinate index of the correlated images (*ζ* − *γ* = *h*), *I_n_*(*x_i_*,*y_j_*)—intensity of the image at the point with *x_i_*, *y_j_*, coordinates recorded for the loading *n* stage and *I_m_*(*x_i_*,*y_j_*)—intensity of the image at the point with *x_i_*, *y_j_*, coordinates recorded in the loading *m* stage.

Next, the lowest value of the *S* sum is searched for by iterative methods. This is carried out by displacement of a selected area of the image recorded for the *n* stage of loading so that it can be found in the image in the *m* stage of loading. Eventually, an image in the form of relative displacements is obtained for particular areas of the recorded image. A scheme of the method operation and selected recorded images from the crack observation are presented in Figure 4.

### 2.4. Description of Experimental Tests

According to the research plan, the determination of the static properties was followed by determination of the mechanical characteristics to be used for a description of the Al–Ti layered material and the fracture toughness of its base materials. The fracture toughness for the linear-elastic range was determined using compact (CT) specimens with the dimension ratio *W*/*B* = 4 (Figure 5) for both ambient (293 K) and cryogenic (77 K) temperature. The application of specimens with the geometry resulting from the above-mentioned ratio was necessary as the target thickness of the sheet was about 10 mm. It was also indirectly determined by technological factors.

The use of such a low temperature to conduct the tests was dictated by the potential application of the tested material. Aviation structures and particularly the aerospace ones for which the analyzed material was developed, are exposed to the impact of low temperatures. Due to the fact that it is difficult to anticipate the impact of differences in the thermal expansion of the base materials on the Al–Ti material strength, it was decided to conduct experiments under cryogenic conditions.

In contrast to the determination of static properties, tests of the fracture toughness in the linear- elastic range consisted of three stages. The first of which involved generating a fatigue crack whose length fitted in the normative range [32]. Due to the fact that each material has different mechanical characteristics, each of them was loaded with a different force during the fatigue crack generation. The loads that were applied to particular materials during the generation of the fatigue crack are presented in Table 2. As mentioned earlier, the length of the fatigue crack was digitally adjusted. Due to the fact that it is not possible to place cameras in liquid nitrogen, the generation of fatigue crack took place under ambient conditions.

The second involving carrying out a monotonic tensile test of the CT specimen with the above-discussed fatigue crack. In the second stage of the tests the speed of the machine piston stroke was 0.02 mm·s^−1^. The process of tension was carried on until the loss of coherence between the stretched parts of the specimen. Since during the second stage of the experiment only the load and crack opening displacement were measured, it was possible to carry out the experiments under both ambient and cryogenic conditions. This was done for all three materials.

The third stage involved determining the geometric features of a crack generated in the tested specimens. Commercially available programs were used for the analysis of the image to enable the determination of the basic planimetric values. Figure 6 presents a graphic form of the testing procedure. It must be noted that optical methods based on the image digital correlation, discussed in Section 3, were used to support the generation of a fatigue crack. As a result, the fatigue cracks of each specimen were provided with very similar lengths. The layered material was a special case because the crack propagated at a different speed on each of its sides. This made it necessary to record and analyze the image on both sides of the specimen. In the considered case, the generation of a fatigue crack was stopped when the crack length on any side of the layered material reached a given value. At least 3 samples for each of the materials have been tested for each temperature condition. The test resulted in 18 curves: load–COD.

Consistently with the basic assumptions of fracture mechanics, cracks can never be equal. Thus, following the experiment, it is necessary to measure the crack length of the CT specimens. In this study this measurement was performed using optical methods.

## 3. Test Results

### 3.1. AA2519 Aluminum Alloy

Diagrams of the curves for load-crack opening determined during tests of the maximal experimental value of the stress intensity factor for heat-treated AA2519 aluminum alloy obtained from the experimental tests are presented in Figure 7. The figure shows diagrams for both temperatures. The results of the tests provided on the basis of the diagrams are included in Table 3.

### 3.2. Ti6Al4V Titanium Alloy

The diagrams obtained as a result of experiments with the use of Ti6Al4V titanium alloy were similar to those obtained for the AA2519 aluminum alloys. Thus, the diagrams for the analyzed Ti6Al4V titanium alloy are presented in Figure 8. Selected results obtained during the described experiment are presented in Table 4.

### 3.3. Layered Material AA2519–AA1050–Ti6Al4V

The maximal experimental value of the stress intensity factor and the values characteristic of the described materials were also determined for the Al–Ti material. As in the case of the base materials, the tests were conducted for two temperature conditions, that is, ambient and cryogenic temperatures. A number of simplifications related to the quantities that refer to the mechanical properties of the AA2519–AA1050–Ti6Al4V composite were used for the description of the layered material. The above-mentioned characteristics were analyzed and adopted for a description of the individual materials, not the composites. When analyzing the results, it must be remembered that in the considered case the tests covered not only two materials placed near each other but also a “structure” which, apart from materials with significantly differing mechanical properties, contains a layer where the materials interact with each other.

This was one of the reasons why the maximal experimental value of the stress intensity factor K_Q_ was assumed to be a comparative quantity, regardless of whether it meets the conditions that qualify it to be a critical value. The diagrams so achieved for the considered Al–Ti are presented in Figure 9 and the test results are presented in Table 5.

It should be noted that the details of determining individual mechanical properties can be found in Boroński et al. [24].

## 4. Analytic Model

### 4.1. Assumptions of the Analytic Model

The possibility of determining the mechanical properties of a layered material by using the information about the mechanical properties of the base materialsis advantageous. In other words, on the basis of the experimental results determined for the base materials, there is a possibility of analytical determination of selected properties for the layered material.

To enable such a solution, it is necessary to develop an analytical model based on e.g., based on a simplified analysis of the nominal stresses of the CT specimens. However, due to the uncertainty of the analytically determined results, it is necessary to an scientific experiments in the field of solid mechanics.

To develop the described analytical model, certain assumptions were made. The primary assumption was that the force loading the Al–Ti layered material is a sum of the partial forces loading the Al and Ti layers:(4)$$P_{Al-Ti}=P_{Al}+P_{Ti}$$

In order to remain independent of the differences associated with different lengths of the fatigue crack, the input values were assumed to refer to real cross-sections of CT specimens, hence further analysis based on the nominal stresses was performed.

According to the scheme shown in Figure 10, a CT specimen undergoes eccentric tension which allows us to determine the value of the maximal nominal stresses of its cross-section as a sum of bending and tension, according to the following Dependence:(5)$$S=\frac{M_{g_{x}}}{W_{x}}+\frac{P}{F}$$
where:(6)$$W_{x}=\;\frac{B\cdot b_{0}^{2}}{6}$$
where *M_g_*—bending moment, *W_x_*—axial index of the cross-section fracture toughness and *F*—uncracked surface area.

The value of the maximal nominal normal stresses in the direction of CT specimen loading can be calculated including the specimen geometric dimensions and its crack length, according to the following Dependence:(7)$$S_{max}=\frac{P\cdot l\cdot 6}{B\cdot b_{0}}+\frac{P}{F}$$
and finally:(8)$$S_{max}=P\left(\frac{6l}{B\cdot b_{0}{}^{2}}+\frac{1}{F}\right)$$

Using Dependence (8) and the experimental results for *P*, the diagram curves were determined for nominal stresses S_max_ as a function of the crack opening COD. The measured crack lengths were used for calculations of the considered nominal stresses. Averaged values of stress diagrams for both base materials are shown in Figure 11 for aluminum alloy and Figure 12 for titanium alloy, respectively. It needs to be noted that the thickest line is used for the diagram curve averaged value calculated from the remaining values.
(9)$$P_{Al}=\frac{S_{Al}}{\frac{6l}{B\cdot b_{0}^{2}}+\frac{1}{F_{Al}}}$$
(10)$$P_{Ti}=\frac{S_{Ti}}{\frac{6l}{B\cdot b_{0}^{2}}+\frac{1}{F_{Ti}}}$$

In order to calculate the loads in the layers of Al and Ti, the values of the nominal stresses for the same COD values were read from S-COD diagrams determined for both base materials.

To enable comparison of the analytic results with the experimental ones, the averaged widths of layers and averaged lengths of cracks measured in real specimens were used for loads *P_Al_* and *P_Ti_* (Figure 10). The quantities that were used for the calculations are presented in Table 6, and a scheme of the determination of the hypothetical load in the layered material, determined on the basis of the nominal stresses of the base materials, is presented in Figure 13.

Further analysis of the diagrams made it possible to determine the parameters to be used to describe the theoretical maximal experimental value of the layered material and compare this with the results of the experimental tests.

### 4.2. Ambient Conditions

A comparison of the diagram of load-crack opening, obtained from calculations with a selected course obtained from the experiment, is shown in Figure 14.

However, a comparison of selected experimentally obtained properties with the theoretical ones is shown in Table 7, and a graphic presentation of the differences normalized to 1 and as a percentage is included in Figure 15 and Figure 16.

The K_IC_ concept was not used in the analysis as a measure of fracture toughness but the K_Q_ concept was used in the model specimen due to the value of the P_max_/P_Q_ quotient. The proposed model re-estimates all the presented values characterizing the material’s brittle fracture toughness. The highest difference was found for the maximal value of force P_max_ and was almost 32%. In the model, the value which is the most significant from the perspective of this study, that is, K_Q_, differed from the experimentally determined one by 19%. The remaining values differed from each other also by about 20%.

For ambient conditions, the presented model does not allow us to determine an accurate value for the layered material fracture toughness based on the properties of the base materials. It only enables an initial estimation of the sought values.

An important factor which can affect the accuracy of the method is the fact that the surface areas of the CT specimen fracture are diverse and they must be averaged to be used in the model. This applies to both the determination of the base diagrams for the layers of Al and Ti, and the force calculations for the theoretical crack length.

### 4.3. Cryogenic Conditions

As in the previous case, a theoretical diagram of load-crack opening was developed for determination of the critical value of the stress intensity factor on the basis of the diagram curves obtained for the base materials. In the analyzed case, a temporary value of K_Q_ was used as a characteristic for comparison of the layered material’s fracture toughness. The above-mentioned diagram curve for cryogenic conditions is shown in Figure 17 and the analysis results are presented in Table 8. Additionally, a graphic presentation of results is shown in Figure 18 and differences between the values measured empirically and those obtained from calculations by means of the proposed model can be found in Figure 19.

The results obtained from calculations performed with the use of the presented model are also burdened with error. The biggest difference occurred for load *P_max_* and was almost 30%. The difference in the remaining values was lower.

As for the *P_Q_* load, the error was slightly over 16%. The model value of *K_Q_* was almost identical to that determined experimentally. As in the previous case, the accuracy of this method depends on the repeatability of the specimen fracture dimensions. In cryogenic conditions the accuracy of the simulation results can additionally be affected by a difference in the thermal expansion of the base materials.

## 5. Conclusions

(1)As regards the experimental value of the stress intensity factor, when determined for Al–Ti layered material, it was not lower than for AA2519 aluminum alloy, for which this value was the lowest.(2)Cryogenic conditions caused a slight drop in the brittle fracture toughness of the Al–Ti layered material and Ti6Al4V titanium alloy. AA2519 aluminum alloy in the linear–elastic range did not react to the considered conditions.(3)The results obtained from the proposed analytic model differ from those obtained experimentally, while the differences are lower for the analysis results obtained for cryogenic conditions.(4)When comparing the results of experimental and analytic analyses, it can be said that the test results obtained using the proposed analytic model are overestimated in almost every case.(5)The main disadvantage of the proposed analytic model is that it neglects the middle layer of AA1050 aluminum alloy situated between the base materials.(6)The inconsistency of the experimental results with the analytic ones may be caused by the joining zone between the welded materials. This is due to a reaction that occurs in the interlayer zone which is difficult to describe. An attempt to provide a description of the interaction can set a new direction of research to be continued in order to get to know the exact mechanical characteristics of layered materials.(7)Despite the differences between the results obtained experimentally and analytically, the proposed model can find application in the initial assessment of the fracture toughness of layered materials built on the basis of AA2519 aluminum alloy and Ti6Al4V titanium alloy.

## Nomenclature

*K_Q_*—stress-intensity factor

*V_k_*—speed of the flame front during detonation

*V_c_*—speed of joining the base materials

*W*—nominal sample width

*B*, *B_e_*—sample thickness

*H**—distance between the horizontal axis of symmetry of the sample and the axis of the clamping slots

*a*—crack length

*b_0_*—uncracked specimen fragment in the crack plane

*α*, *β*, *γ*, *ζ*—values of the coordinates of indexed images

*P*, *P_max_*, *P_Q_*—load

*M_g_*—bending moment

*W_x_*—axial index of the cross-section fracture toughness

*F*—uncracked surface area

*l*—the distance between the force application point (pin axis) and the back of the sample

## Figures and Tables

**Figure 1 materials-13-04439-f001:**
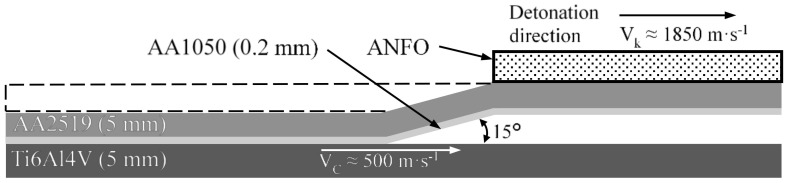
Schematic presentation of the process of AA2519–AA1050–Ti6Al4V composite creation.

**Figure 2 materials-13-04439-f002:**
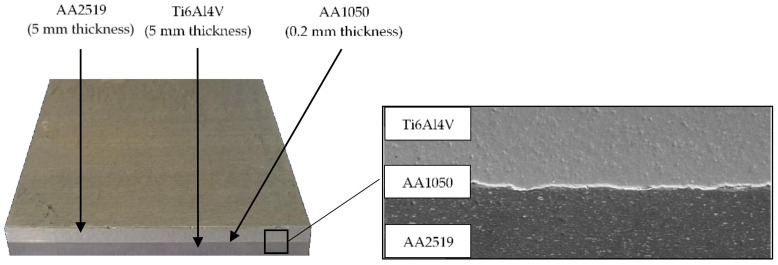
Layered material AA2519–AA1050–Ti6Al4V.

**Figure 3 materials-13-04439-f003:**
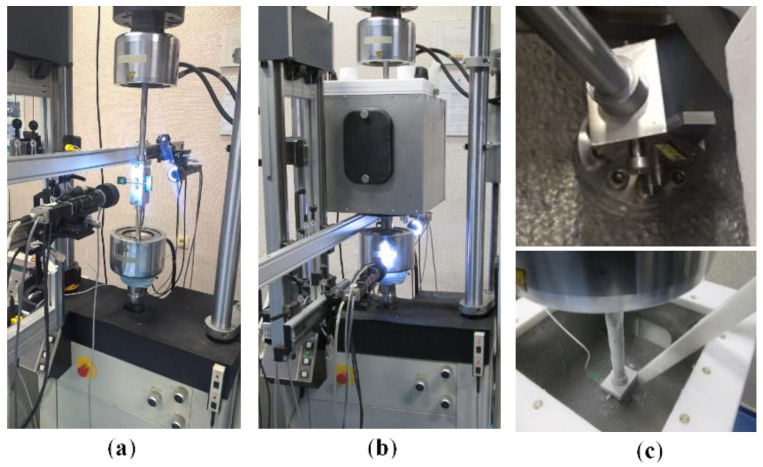
Test stand: (**a**) stand for generation of fatigue crack and tests at ambient temperature; (**b**) stand with an environmental chamber and (**c**) view inside the chamber.

**Figure 4 materials-13-04439-f004:**
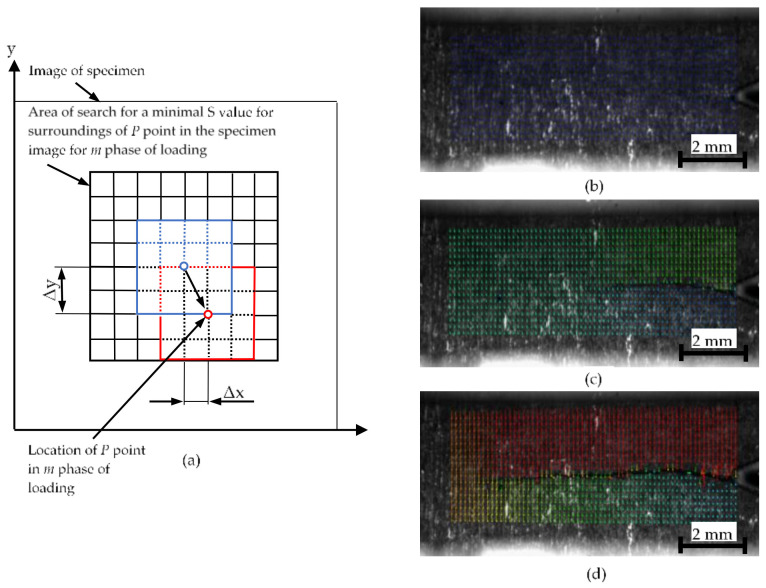
Analysis of the specimen image with the use of a digital correlation: (**a**) scheme of operation; (**b**–**d**) particular stages of the crack propagation.

**Figure 5 materials-13-04439-f005:**
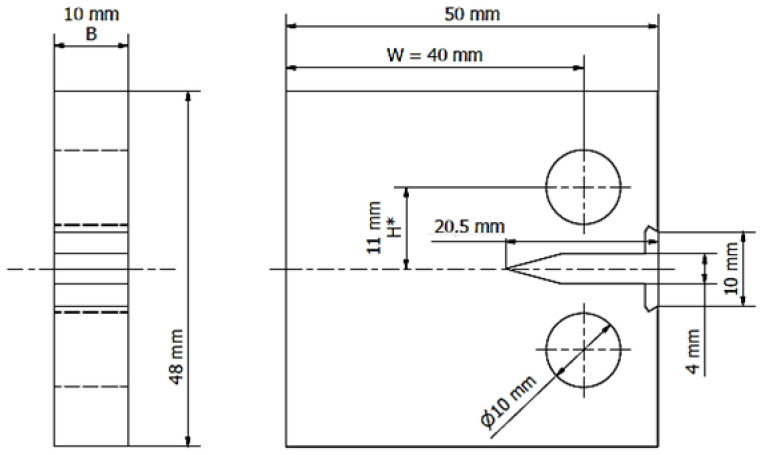
Basic dimensions of CT specimen used for determination of mechanical characteristics describing fracture toughness of Al–Ti layered material and its base materials.

**Figure 6 materials-13-04439-f006:**
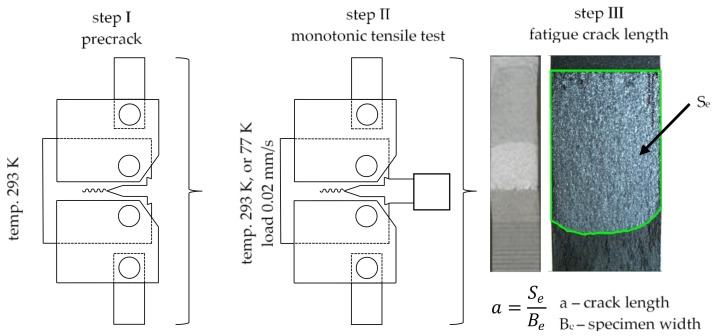
Graphic presentation of the testing procedure.

**Figure 7 materials-13-04439-f007:**
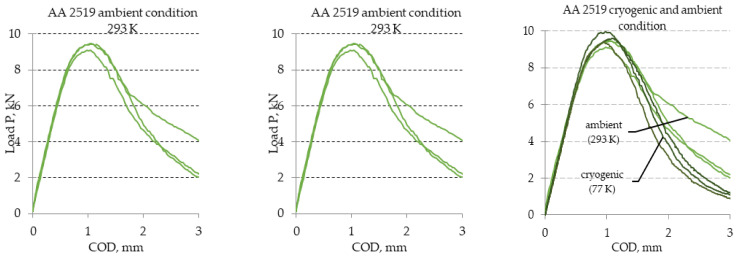
Diagram of load-COD determined during tests of maximal experimental value of stress intensity factor K_Q_ determined for aluminium alloy AA2519 under ambient and cryogenic conditions.

**Figure 8 materials-13-04439-f008:**
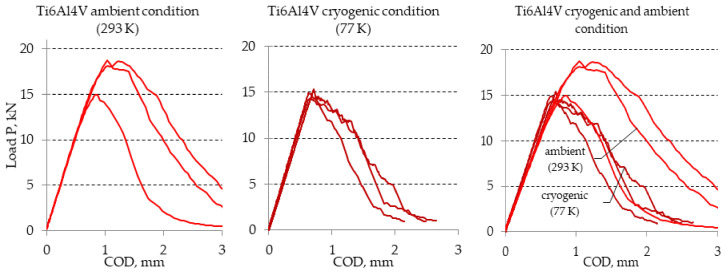
Diagrams of load-COD determined during tests of maximal experimental value of stress intensity factor K_Q_ determined for titanium alloy Ti6Al4V under ambient and cryogenic conditions.

**Figure 9 materials-13-04439-f009:**
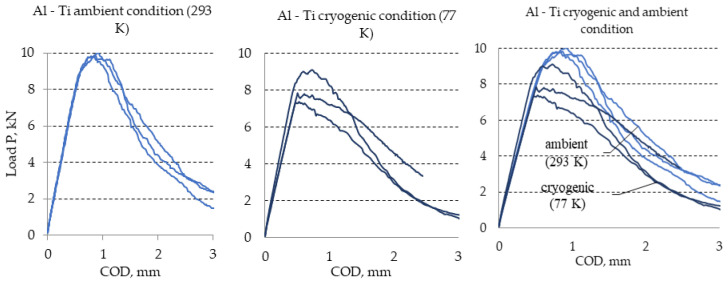
Diagrams of load–COD determined during the tests of the maximal experimental value of stress intensity factor K_Q_ determined for the layered material AA2519–AA1050–Ti6Al4V under ambient and cryogenic conditions.

**Figure 10 materials-13-04439-f010:**
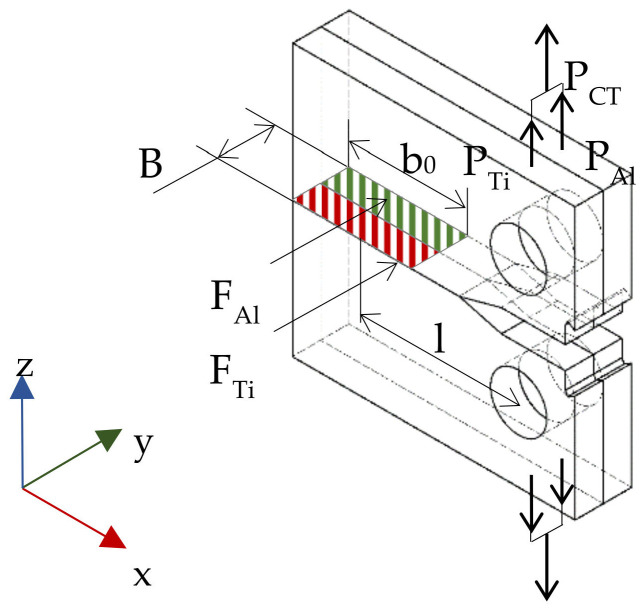
Scheme of the model reference system.

**Figure 11 materials-13-04439-f011:**
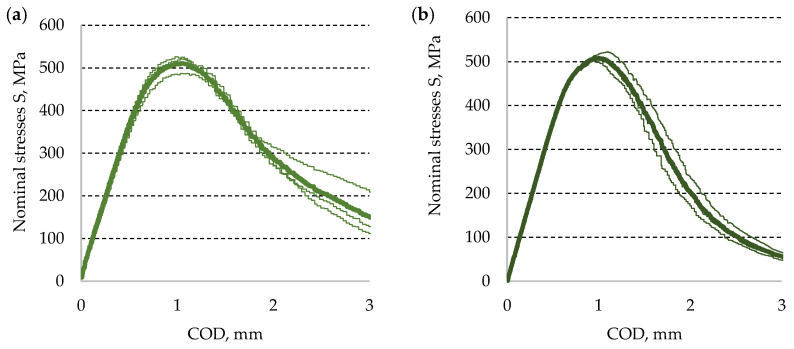
Diagrams of nominal stresses–specimen opening, calculated for AA2519 aluminum alloy: (**a**) ambient conditions and (**b**) cryogenic conditions.

**Figure 12 materials-13-04439-f012:**
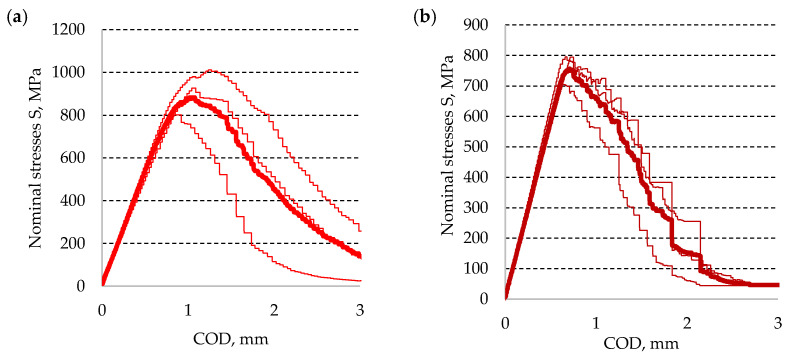
Diagrams of nominal stresses–specimen opening, calculated for Ti6Al4V titanium alloy: (**a**) ambient conditions and (**b**) cryogenic conditions.

**Figure 13 materials-13-04439-f013:**
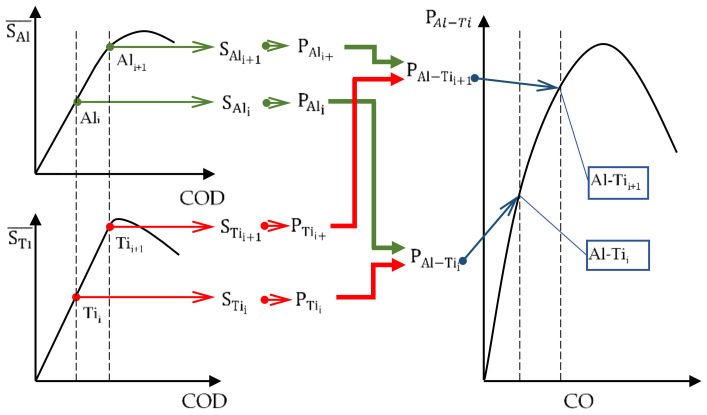
Scheme of determination of hypothetical load occurring in a layered material calculated on the basis of nominal stresses in specimens made of base materials.

**Figure 14 materials-13-04439-f014:**
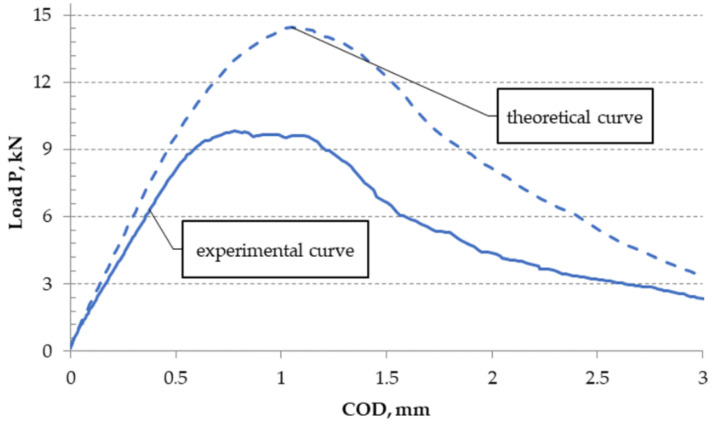
Theoretical and selected experimental diagram curves of the load–crack opening displacement obtained from calculations for a layered Al–Ti material for ambient conditions.

**Figure 15 materials-13-04439-f015:**
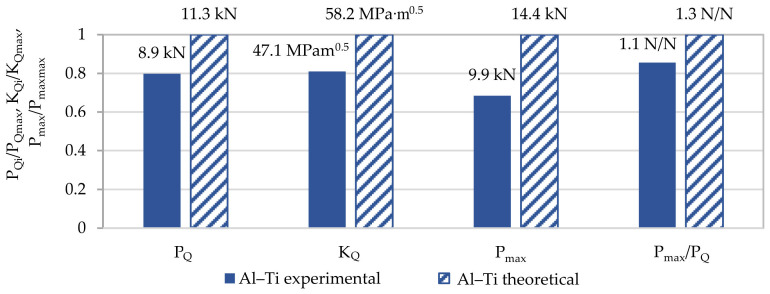
Normalized to 1 comparison of selected mechanical properties obtained from experiment and calculations characterizing fracture toughness of Al–Ti layered material in ambient conditions.

**Figure 16 materials-13-04439-f016:**
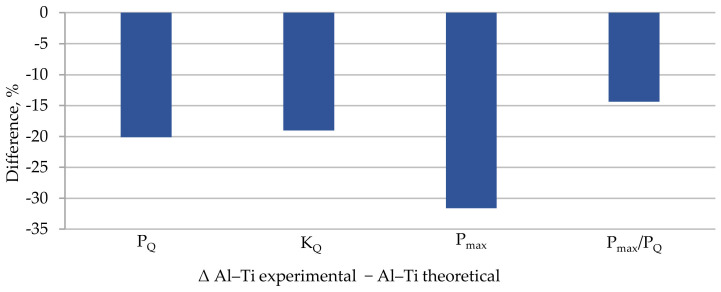
Differences between selected mechanical properties obtained from experiment and calculations for Al–Ti layered material in ambient conditions.

**Figure 17 materials-13-04439-f017:**
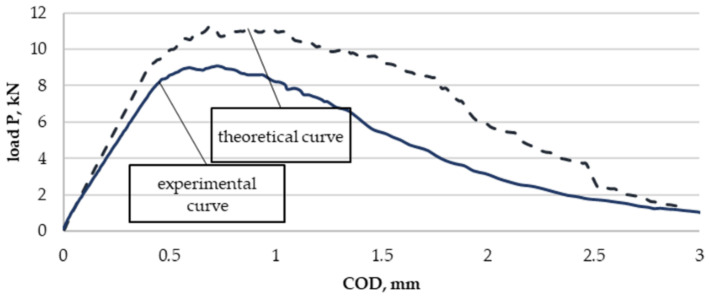
Theoretical and selected experimental curves of load–crack opening obtained from calculations for Al–Ti layered material for cryogenic conditions.

**Figure 18 materials-13-04439-f018:**
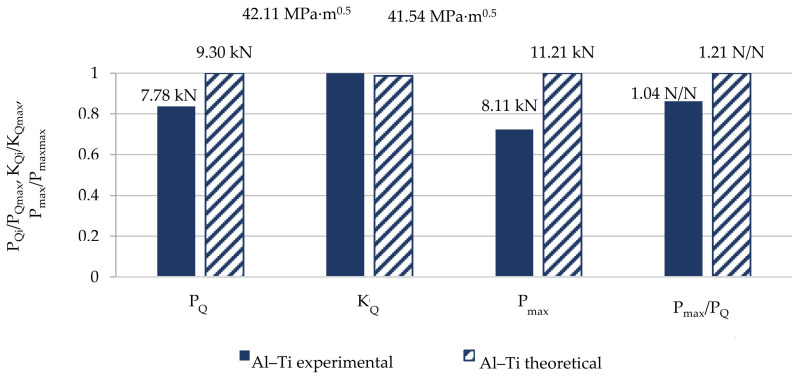
Normalized to 1 comparison of selected mechanical properties obtained experimentally and theoretically characterizing fracture toughness of, Al–Ti material for cryogenic conditions.

**Figure 19 materials-13-04439-f019:**
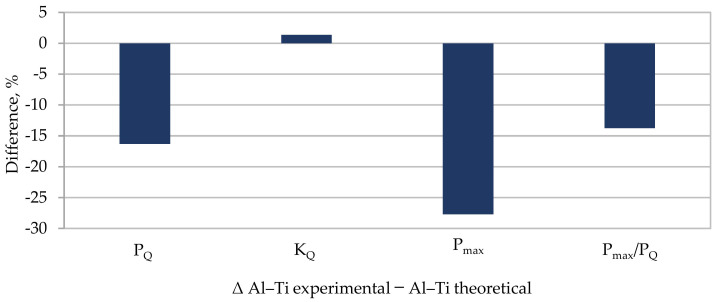
Differences between selected mechanical properties obtained experimentally and theoretically, for Al–Ti layered material for cryogenic conditions.

**Table 1 materials-13-04439-t001:** Chemical composition of materials (wt.%) that make up AA2519–AA1050–Ti6Al4V layered material [17,25,31].

Materials	Chemical Composition, % Mass						
AA 2519	Si	Fe	Cu	Mg	Zn	Ti	V
	0.06	0.08	5.77	0.18	0.01	0.04	0.12
AA 1050	Mn	Fe	Si	Mg	Cu	Ti	Zn
	0.05	0,4	0.25	0.05	0.06	0.05	0.07
Ti6Al4V	Al	V	Fe	Si	O	C	N
	6.42	4.12	0.18	0.024	0.12	0.013	0.011

**Table 2 materials-13-04439-t002:** Presentation of set parameters used in generation of a fatigue crack.

No.	Material	Frequency (Hz)	Loading (kN)
1	AA2519	5	4.7
2	Ti6Al4V	5	6.8
3	Al–Ti	5	5.8

**Table 3 materials-13-04439-t003:** Results of fracture toughness tests in a linear-elastic range for AA2519 aluminum alloy under cryogenic conditions.

AA2519		PQ kN	KQ MPa·m^0.5^	Pmax kN	Pmax/PQ -
Temperature K	293 K	7.8	40.3	9.3	1.2
	77 K	8.1	40.4	9.7	1.2

**Table 4 materials-13-04439-t004:** Results of fracture toughness tests in a linear-elastic range of Ti6Al4V titanium under cryogenic and ambient conditions.

Ti6Al4V		P_Q_ kN	K_Q_ MPa·m^0.5^	P_max_ kN	P_max_/P_Q_ -
Temperature K	293 K	15.2	74.6	17.5	1.1
	77 K	14.7	70.7	14.9	1.0

**Table 5 materials-13-04439-t005:** Results of fracture toughness tests in a linear-elastic range for AA2519–AA1050–Ti6Al4V layered material determined for cryogenic conditions.

AA2519–AA1050–Ti6Al4V		P_Q_, kN	K_Q_, MPa·m^0.5^	P_max_, kN	P_max_/P_Q_
Temperature, K	293 K	9.0	47.1	9.9	1.1
	77 K	7.8	42.1	8.1	1.0

**Table 6 materials-13-04439-t006:** Averaged values of the width and length of fatigue cracks of base materials of the layered material.

Description	Width B	*a_w_*	Width Al	Width Ti	Fracture Al	Fracture Ti
Average value	9.513	8.476	4.853	4.660	9.108	7.843
Standard deviation	0.020	0.728	0.062	0.065	1.054	1.016

**Table 7 materials-13-04439-t007:** Comparison of selected mechanical properties obtained experimentally and through calculations for the layered Al–Ti material determined in ambient conditions.

Denotation	P_Q_kN	K_Q_MPa·m^0,5^	P_max_kN	P_max_/P_Q_-
Al–Ti Theoretical (W)	9.0	47.1	9.9	1.1
Al–Ti Theoretical (N)	11.3	58.2	14.4	1.3
Comparison %	difference *ΔX* = ((*X_W_* − *X_N_*)/*X_W_*) × 100%, where *X*: *P_Q_*, *K_Q_*, *P_max_*, *P_max_*/*P_Q_*			
	−20.1	−19.0	−31.6	−14.4

**Table 8 materials-13-04439-t008:** Comparison of selected mechanical properties obtained experimentally and theoretically for Al–Ti layered material for cryogenic conditions.

Denotation	P_Q_kN	K_Q_MPa·m^0.5^	P_max_kN	P_max_/P_Q_-
Al–Ti experimental (W)	7.78	42.11	8.11	1.04
Al–Ti theoretical (N)	9.30	41.54	11.21	1.21
Comparison %	difference Δ*X* = ((*X_W_ − X_N_*)/*X_W_*) × 100%, where *X*: *P_Q_*, *K_Q_*, *P_max_*, *P_max_*/*P_Q_*			
	−16.31	1.38	−27.70	−13.77

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
