# Peer review of "Analytic Model of Maximal Experimental Value of Stress Intensity Factor KQ for AA2519–AA1050–Ti6Al4V Layered Material"

_materials, 2020, doi:10.3390/ma13194439_

Round 1

Reviewer 1 Report

General Evaluation

This is an interesting and well written paper concerning the fracture mechanics properties of explosively welded sandwich composites (AA2519-AA1050-Ti6Al4V). In the present study the fracture mechanics parameter is calculated by the individual components of the composite (AA2519 and Ti6Al4V) using force balance equations and average crack dimensions. The analytical results are compared with the experimental ones. The paper is within the scope of the Journal and could be accepted for publication provided some comments and suggestions.

Technical/Scientific Comments

  1. Due to the high number of examined parameters, a list of symbols is recommended to be inserted.
  2. In Table 1, please add the composition units (wt.%).
  3. Line 100: 2 mm (should be 0.2 mm according to Figure 1).
  4. Please describe the microstructure and the bonding quality of the explosively welded component. If there are representative micrographs, it is recommended to be added.
  5. In the caption of Figure 7-9, triplicate specimens should be mentioned, for each testing temperature.
  6. Please include the basic fracture mechanics equation for the determination of KQ.
  7. In Table 3, the number of decimal digits of KQ should be reduced. Also, the load ratio is an arithmetic quantity (the N/N in units should be eliminated). The same comment is valid for Tables 4 and 5.
  8. Please analyze further the parameter Wx in Eq. (3) and how the Eq. (4) was deduced.
  9. In Table 6, there are identical names of columns referred to the width value. Please explain.
  10. In Tables 7 and 8, please check the numbers. They might have been erroneously completed. See for instance the PQ (Table 7) and Pmax (Table 8) for the computational curve and the difference for the load ratio. Check also that the difference should be read in %. (Use either theoretical, or analytical or computational as one single term throughout the text to avoid confusion)
  11. The addition of representative fractographs of the specimens could be complementary raising the completeness of the study.

Language/Grammatical comments

The language is in general sufficient. However, a final proofreading is suggested to eliminate typographical and grammatical errors.

Author Response

Thank you very much for reviewing my paper. My answers are in the file. Please read it.
Thank you again!

Reviewer 2 Report

This paper succeeds in the evaluation of stress intensity factor and its confirmation. I would like to recommend its publication after a minor revision. I suggest that the author should merge some relevant paragraphs.

Author Response

(The authors gave the same response as above.)

Reviewer 3 Report

Please find attached a revised version of your manuscript with comments, questions and suggestions. Generally, much more discussion of the results is needed. For instance, the correlation between the analytical model and the experimental results is not well described. Moreover, the way the results were obtained or measured is not mentioned. 

I hope the suggestions will help improve the clarity of the work. 

Good luck! 

Author Response

(The authors gave the same response as above.)
